# Propensity Score Matching Underestimates Real Treatment Effect, in a Simulated Theoretical Multivariate Model

Daniel Garcia Iglesias [1,2]

1    Arrhythmia Unit, Cardiology Department, Hospital Universitario Central de Asturias, 33011 Oviedo, Spain; daniel.garciai@sespa.es; Tel.: +34-985280000
2    Instituto de Investigación Sanitaria del Principado de Asturias, 33011 Oviedo, Spain

**Abstract:** Propensity Score Matching (PSM) is a useful method to reduce the impact of Treatment-Selection Bias in the estimation of causal effects in observational studies. After matching, the PSM significantly reduces the sample under investigation, which may lead to other possible biases (due to overfitting, excess of covariation or a reduced number of observations). In this sense, we wanted to analyze the behavior of this PSM compared with other widely used methods to deal with non-comparable groups, such as the Multivariate Regression Model (MRM). Monte Carlo Simulations are made to construct groups with different effects in order to compare the behavior of PSM and MRM estimating these effects. In addition, the Treatment Selection Bias reduction for the PSM is calculated. With the PSM a reduction in the Treatment Selection Bias is achieved (0.983 [0.982, 0.984]), with a reduction in the Relative Real Treatment Effect Estimation Error (0.216 [0.2, 0.232]), but despite this bias reduction and estimation error reduction, the MRM reduces this estimation error significantly more than the PSM (0.539 [0.522, 0.556], $p < 0.001$). In addition, the PSM leads to a 30% reduction in the sample. This loss of information derived from the matching process may lead to another not known bias and thus to the inaccuracy of the effect estimation compared with the MRM.

**Keywords:** propensity score matching; multivariate analysis; general linear model; Monte Carlo method; causal effect estimation; observational study

## 1. Introduction

Propensity Score Matching (PSM) is a useful method to reduce the impact of treatment selection bias in the estimation of causal effects in observational studies. Since firstly described by Rosenbaum and Rubin in 1983 [1], its utility in Medicine, Psychology, Economics and other fields has increased exponentially in the last years, reaching a 17-fold increase in recent years [2,3]. Although it does not bypass the necessity for randomized studies, it may be an alternative to reduce the impact of treatment selection bias in observational studies.

The Propensity Score (PS) is defined as the subject's probability of receiving a specific treatment conditional on the observed covariates [1]. After stratification by its PS, treated and untreated patients are matched by their PS with the most similar individuals of the opposite group. It leads to a more similar distribution of baseline characteristics between treated and untreated subjects, and it has been demonstrated that because of this more homogeneous distribution of basal characteristics, this method reduces the treatment selection bias [4,5].

Once two comparable groups have been obtained, researchers treat PSM studies more similarly to randomized studies (although it does not substitute this randomized studies) and use them as a reasonable alternative for observational studies [6]. In this sense, it is thought that because PSM controls the possible treatment selection bias, it would be possible to directly measure the effects in both matched groups, and thus it may be better for observational studies than other multivariate adjustment methods.

One important concern is the influence of the significant loss of non-matched individuals that may be seen in some works using this PSM method [7] and who are not used for posterior analysis. Because we need to eliminate enough unmatched individuals to guarantee in some way the Treatment Selection Bias correction, it is not possible to know if the elimination of this unmatched individuals can cause some loss of information that in other ways would be analyzed and therefore lead us to a non controlled bias. In this sense other authors have reported the over-employment of this technique and its potential implications in potential biases [8,9].

Moreover, although we are controlling the Treatment Selection Bias, after matching, we are directly measuring the effect in both matched groups. In this way, compared with a multivariate adjustment method, which is widely used to control groups for other possible confounders, may lead to possible errors in effect estimation (due to overfitting, excess of covariation or reduced number of observations) [10–12]. Despite this possible errors, as we previously mentioned, its utility has increased exponentially over the last years, so we need to address special awareness when using this methods. Because of that, we wanted to test the behavior of PSM in different situations, compared with a multivariate analysis based on General Linear Models (GLM) in the estimation of treatment effects to highlight the possible errors of this technique compared with a multivariate adjustment method. For this purpose, we developed a theoretical Monte Carlo Method of treatment effects in which we applied the PSM and a multivariate analysis based on a GLM to compare their ability to estimate the Real Treatment Effect (RTE) in each situation.

## 2. Materials and Methods

### 2.1. Theoretical Multivariate Model

Suppose $Y_z(x)$ is the patient $z$ probability for a certain event. Its probability may be influenced by a series of independent variables $A_{zj}$ and $B_{zk}$, each one of them with a concrete weight in this patient probability prediction $q_j$ and $s_k$, respectively. $A_{zj}$ variables may be related to the received treatment, and $B_{zk}$ variables are supposed to be independent of the received treatment. It may be also influenced by the treatment status $X_z$ of the patient, which may confer some protection $t$ against the event under study.

For each patient, there may be also some unmeasured influence of $\epsilon_z$ in its event probability, which may vary from patient to patient and may be because some unknown or non-measured variables. We consider it as a random variable, which may be based in a normal distribution:

$$\epsilon_z \sim \mathcal{N}(0, 1) \tag{1}$$

The unmeasured influence in the probability $\epsilon$ and the $t$ and $s_1, \ldots, s_k$ weights will have positive values if predispose to the adverse event, and thus, negative values are protectors in some way to the adverse event.

Each patient may present some different characteristics. A part from these $A_{zj}$ and $B_{zk}$ characteristics that may predispose in some way the probability for the event under study, it may present other variables unrelated with the event under study. Some of them may predispose to receive the treatment under study $C_l$, and others, $D_m$, may be unrelated to either patient outcomes or treatment predisposition.

For a concrete patient, the logit of the probability for a certain event may be predicted by the formula:

$$Y_z(x) = X_z t + A_{z1} q_1 + \ldots + A_{zj} q_j + B_{z1} s_1 + \ldots + B_{zk} s_k + \epsilon_z \tag{2}$$

Thus, the theoretical event probability for a $z$ given patient will be:

$$\rho_z = \frac{e^{Y_z(x)}}{1 + e^{Y_z(x)}} \tag{3}$$

For posterior metrics, the RTE will be the estimated coefficient [13]. This RTE will be defined as the Odds Ratio of this probability:

$$RTE = e^{Y_z(x)} \tag{4}$$

### 2.2. Monte Carlo Simulations

Once the theoretical model was built, Monte Carlo Simulations were made to construct groups for a posterior analysis. Simulated experiments under this conditions were made, and an event status $\mathbb{E}_z$ was assigned to each patient $z$ in each simulated experiment, based on a binomial distribution with $n = 1$ and probability $p = \rho_z$:

$$\mathbb{E}_z \sim \mathcal{B}(1, \rho_z) \tag{5}$$

The theoretical (real) treatment effect was modified after each group of simulations, ranging from a 0 (null) effect to a 5-fold event reduction $(-5)$ in 0.1 intervals.

Apart from the treatment status and the treatment effect in each simulation, the previously described groups of variables with each concrete weight were introduced in the model for each simulation. This variables were distributed by a binomial distribution with a probability that varied depending on its correlation with other variables. In each group of variables, each variable weight ranged from $-2$ (protects) to 2 (predisposes).

A total of $5 \times 10^6$ Monte Carlo Simulations were conducted. There were 50 blocks of experiments, each one with a different real treatment effect. In each block of experiments, each experiment was simulated with 500 individuals and later repeated 200 times for each real treatment effect.

### 2.3. Unadjusted Model

The Odds Ratio for the event prevention under Treatment status was calculated with a univariable General Linear Model (GLM). For each experiment, with the complete matrix of events $Y$ and treatment $X$ status, a GLM was built to estimate the unadjusted estimated risk prevention effect for the treatment $t_{ua}$ ($ua$ stands for Unadjusted Model):

$$Y_{ua} = X t_{ua} + U \tag{6}$$

where $U$ is the error derived from the effects not measured by the model.

From this built model, the estimated Odds Ratio for the RTE $OR_{ua}$ was calculated:

$$OR_{ua} = e^{t_{ua}} \tag{7}$$

Since this measured Odds Ratio did not take in account other variables, it will be named the unadjusted Odds Ratio, and it will be considered the reference for the improvement in the RTE estimation.

### 2.4. Multivariate Regression Model

For control purposes, a Multivariate Regression Model (MRM) was built for each experiment with all the variables under analysis. In each experiment, the complete matrix of events $Y$, treatment status $X$ and analyzed variables $A$ (outcome predictors related to the received treatment), $B$ (outcome predictors independent of the received treatment), $C$ (variables that predispose to the received treatment) and $D$ (variables unrelated to both outcomes and received treatment) were used to build the MRM and estimate the multivariate adjusted event reduction of the treatment $t_{multi}$ ($multi$ stands for Multivariate Regression Model):

$$Y_{multi} = X t_{multi} + A q_{multi} + B s_{multi} + C u_{multi} + D v_{multi} + U \tag{8}$$

where $q$, $s$, $u$ and $v$ are the estimated weights of each kind of variable by the Multivariate Regression Model.

From this built model, the estimated Odds Ratio for the RTE (Multivariate Odds Ratio, $OR_{multi}$) is calculated:

$$OR_{multi} = e^{t_{multi}} \tag{9}$$

It will be used as the gold standard for the estimation of the RTE.

### 2.5. Propensity Score Matching Model

PSM was used to estimate the Treatment Effect. As described early, the PS was built based on a MRM designed to estimate each patient's predisposition to receive the treatment under investigation. In each patient, the PS was calculated for posterior matching between treated and untreated individuals. For matching, the nearest method was used (with a caliper of 0.2) with the R algorithm included in the MatchIt library [14].

Once matching was completed, the treatment effect was estimated based on a GLM. For each experiment, the matrix of matched individuals with the events $Y_{match}$ and treatment $X_{match}$ status were used to build a GLM to estimate the estimated risk prevention effect for the treatment $t_{match}$ (*match* stands for Propensity Score Matching Model):

$$Y_{match} = X_{match}t_{match} + U \tag{10}$$

From this built model, the estimated Odds Ratio for the RTE $OR_{match}$ was calculated:

$$OR_{match} = e^{t_{match}} \tag{11}$$

### 2.6. Statistical Analysis

For each treatment effect situation, RTE Estimation was measured with each one of the three described methods. This RTE estimation was later compared with the RTE to calculate the inaccuracy of the RTE Estimation (Relative RTE Estimation Error). All variables were expressed as mean $+/-$ 95% Confidence Intervals. The comparison of the RTE Estimation Error between the three described methods was performed with a multivariate analysis of variance (MANOVA test).

For the MRM and the PSM Model, the RTE Estimation Error was compared with the Unadjusted Model to calculate the Relative RTE Estimation Error Reduction for each one. The comparison of this Relative RTE Estimation Error Reduction between MRM and PSM Model was performed with a paired T Test.

For the Unadjusted Model and the PSM Model, the Treatment Selection Bias was calculated with a Pearson's chi-squared test to evaluate the homogeneity between groups. The comparison of the Treatment Selection bias between both models was performed with a paired T Test comparing the the chi-squared test statistic of both models, and the reduction in the Treatment Selection bias with the PSM Model was expressed as mean $+/-$ 95% Confidence Intervals.

For the PSM Model, the percentage of excluded patients in each analysis was also analyzed. This was be expressed as the mean of the percentage exclusions in each analysis and the 95% Confidence Interval of the percentage exclusions in each analysis.

### 2.7. Analysis Software

For the data generation with the Monte Carlo Simulations and the posterior analysis, the open software R [15] was used. The R library MatchIt [14] was used for the PSM. All written code for this purpose is available through the PropensityScoreReview repository [16].

### 3. Results

*3.1. Treatment Selection Bias Reduction with Propensity Score Matching*

The PSM Model significantly reduces the Treatment Selection Bias in all scenarios. As seen in Table 1 and Figure 1, there is a Relative Treatment Selection Bias reduction of about 0.98 in all scenarios. The main problem of this model is the important number of excluded patients (there is only a 70% of patients that are included for the analysis).

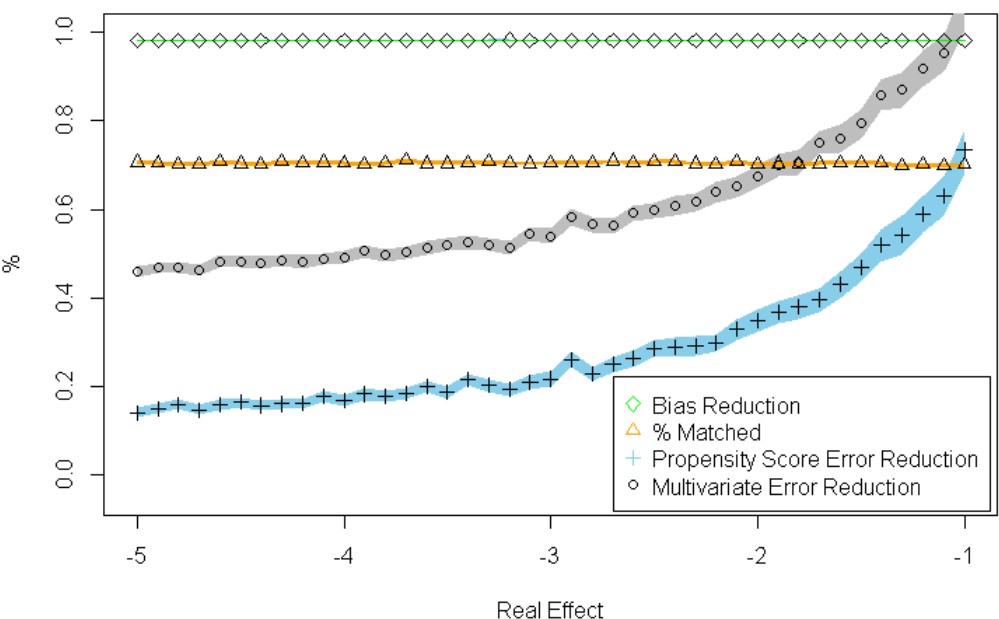

**Figure 1.** Comparison of the Real Treatment Effect Estimation Reduction with the Multivariate Model and the Propensity Score Matching Model.

**Table 1.** Treatment Selection Bias for the Unadjusted Model and the Propensity Score Matching Model. RTE: Real Treatment Effect; T-S Bias: Treatment Selection Bias. In brackets: 95% confidence interval.

| RTE | T-S Bias * | | | Excluded Patients (%) |
|---|---|---|---|---|
| | Unadjusted Model | Matched Model | T-S Bias Reduction | |
| 5 | 220.7 (217.82, 223.57) | 3.631 (3.479, 3.783) | 0.983 (0.983, 0.984) | 70.66 (70.2, 71.12) |
| 4 | 219.09 (216.36, 221.83) | 3.599 (3.446, 3.751) | 0.983 (0.983, 0.984) | 70.43 (69.98, 70.88) |
| 3 | 219.19 (216.29, 221.90) | 3.698 (3.536, 3.861) | 0.983 (0.982, 0.984) | 70.55 (70.1, 71) |
| 2 | 217.6 (214.80, 220.33) | 3.721 (3.576, 3.866) | 0.983 (0.982, 0.983) | 70.12 (69.68, 70.56) |
| 1 | 216.98 (214.31, 219.66) | 3.738 (3.603, 3.873) | 0.982 (0.982, 0.983) | 70.19 (69.72, 70.65) |
| 0 | 219.47 (216.85, 222.09) | 3.684 (3.532, 3.835) | 0.983 (0.982, 0.984) | 70.21 (69.77, 70.65) |

* Tests applied presented statistically significant differences ($p < 0.001$).

*3.2. Real Treatment Effect Estimation and Relative Real Treatment Effect Estimation Error Reduction with Propensity Score Matching Model and Multivariate Regression Model*

As it can be seen in Figures 2 and 3 and Table 2, the PSM Model and the MRM significantly estimate a more accurate RTE than the Unadjusted Model. This PSM Model and MRM present a significantly reduced Relative RTE Estimated Error, compared with the Unadjusted Model.

The MRM also presents a Relative RTE Estimated Error significantly lower than the PSM Model, which leads to a significantly increased Relative RTE Estimated Error Reduction compared with the PSM Model.

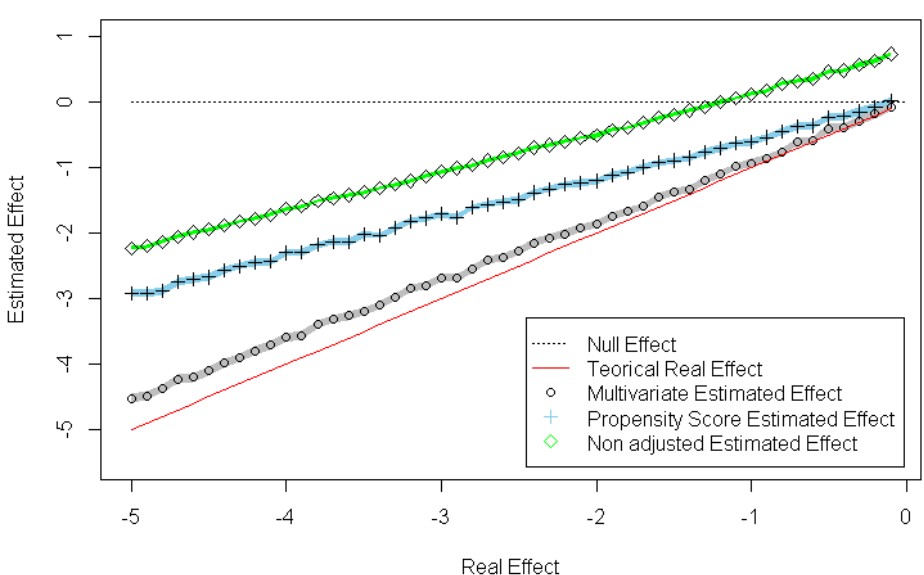

**Figure 2.** Comparison of Real Treatment Effect and Real Treatment Effect Estimation from the Unadjusted Model, Multivariate Regression Model and Propensity Score Matching Model.

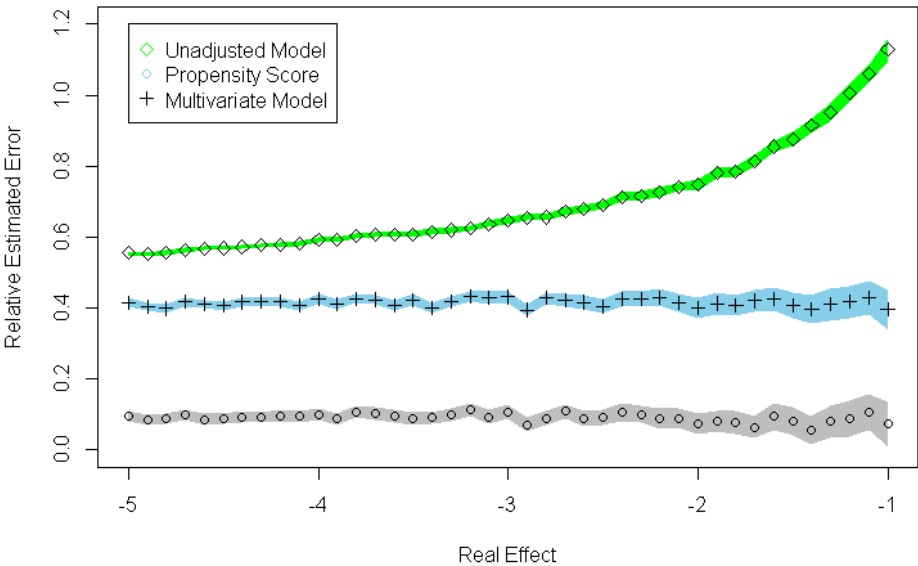

**Figure 3.** Comparison of Relative Real Treatment Effect Estimation Error with the Unadjusted Model, Multivariate Model and Propensity Score Matching Model.

**Table 2.** Real Treatment Effect Estimation for the Unadjusted Multivariate and Propensity Score Matching models. RTE: Real Treatment Effect Estimation; T-S Bias: Treatment Selection Bias. In brackets: 95% confidence interval.

| RTE | Estimated Treatment Effect * | | | Relative RTE Estimated Error * | | | RTE Estimated Error Reduction * | |
|---|---|---|---|---|---|---|---|---|
| | Unadjusted Model | Matched Model | Multivariate Model | Unadjusted Model | Matched Model | Multivariate Model | Matched Model | Multivariate Model |
| 5 | −2.225 (−2.254, −2.195) | −2.922 (−2.984, −2.86) | −4.521 (−4.594, −4.448) | 0.555 (0.549, 0.561) | 0.416 (0.403, 0.428) | 0.096 (0.082, 0.11) | 0.139 (0.128, 0.151) | 0.459 (0.446, 0.473) |
| 4 | −1.63 (−1.659, −1.601) | −2.3 (−2.355, −2.245) | −3.597 (−3.659, −3.536) | 0.592 (0.585, 0.6) | 0.425 (0.411, 0.439) | 0.101 (0.085, 0.116) | 0.167 (0.154, 0.181) | 0.492 (0.478, 0.506) |
| 3 | −1.059 (−1.088, −1.030) | −1.707 (−1.759, −1.655) | −2.676 (−2.736, −2.616) | 0.647 (0.637, 0.657) | 0.431 (0.41, 0.448) | 0.108 (0.088, 0.128) | 0.216 (0.2, 0.232) | 0.539 (0.522, 0.556) |
| 2 | −0.504 (−0.536, −0.472) | −1.201 (−1.257, −1.144) | −1.852 (−1.91, −1.793) | 0.748 (0.732, 0.76) | 0.4 (0.371, 0.428) | 0.074 (0.045, 0.103) | 0.348 (0.324, 0.373) | 0.674 (0.649, 0.698) |
| 1 | 0.129 (−0.096, 0.161) | −0.604 (−0.661, −0.548) | −0.927 (−0.99, −0.864) | 1.129 (1.096, 1.161) | 0.396 (0.339, 0.452) | 0.073 (0.01, 0.136) | 0.733 (0.684, 0.783) | 1.056 (1.003, 1.109) |
| 0 | 0.743 (−0.708, 0.777) | 0.026 (−0.032, 0.085) | −0.078 (−0.139, −0.017) | 8.426 (8.08, 8.772) | 1.264 (0.676, 1.853) | 0.22 (−0.394, 0.83) | 7.162 (6.622, 7.701) | 8.206 (7.699, 8.714) |

\* Tests applied presented statistically significant differences ($p < 0.001$).

## 4. Discussion

PSM has been widely used in different subjects for Treatment Selection Bias reduction. As we show in this present work, PSM corrects the Treatment Selection Bias properly, obtaining two comparable groups, so we can thus directly measure the effect under investigation. In this sense, in our work, the Treatment Selection Bias practically disappears with the PSM (Treatment Selection Bias reduction of 0.982–0.983 among all scenarios). However, despite this reduction, PSM still fails in the RTE estimation, compared with MRM. We demonstrate how despite this reduction in Treatment Selection Bias, the Relative RTE Estimation error rounds to 0.4 while the MRM is 4 times smaller (it rounds to 0.1).

As we mentioned earlier, our main concern about the PSM method is the percentage of unmatched individuals and its possible influence on the posterior estimation of RTE. Other authors have also suggested how the PSM can misestimate the effect compared with MRM [10–12]. In our work, there is significant sample reduction, with a percentage of analyzed individuals of 70.12 to 70.66% from the total individuals under investigation. In addition, it may be the reason for the inaccuracy of the RTE Estimation compared with the MRM.

MRMs do not deal with two comparable populations, but instead, they weigh the different variables under study. Despite dealing with non-comparable groups, thanks to this weighting of the analyzed variables, the MRM can find a solution in a different way than the problem of the Treatment Selection Bias. In addition, since no individual is eliminated from the analysis, there is no loss of information, reducing other potential biases that may appear in the PSM method.

In our present work, although both (PSM and MRM) reduce the Relative RTE Estimation Error, this reduction is better with the MRM than the PSM. This best performance of the MRM may confirm our previous preoccupation about the possible influence of the sample reduction in the posterior estimation of the RTE.

Since there may be a significant reduction in the sample under analysis when we are using a PSM method, we have to take it in account before accepting the obtained results, especially when this reduction is important. Other multivariate methods should always be performed in addition to the PSM analysis and both results compared in order to seek a possible uncontrolled bias. If a significant difference is obtained between both analyses, we have to suspect a possible bias derived from the sample reduction once the matching has been performed.

## 5. Conclusions

With the PSM, a reduction in the Treatment Selection Bias is achieved with a reduction in the Relative Real Treatment Effect Estimation Error, but the MRM reduces this estimation error significantly more compared with the PSM. In addition, the PSM leads to a significant reduction in the sample, with some loss of information derived from the matching process that may lead to another not known bias and thus to the inaccuracy of the effect estimation compared with the MRM.

**Funding:** This research received no external funding.

**Data Availability Statement:** All written code for this manuscript is available through the Github PropensityScoreReview repository.

**Conflicts of Interest:** The author declares no conflict of interest.

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
