# Peer review of "Propensity Score Matching Underestimates Real Treatment Effect, in a Simulated Theoretical Multivariate Model"

_mathematics, doi:10.3390/math10091547_

Round 1

Reviewer 1 Report

I consider the study to be very relevant and articles of this nature (simulations) should be valued, as its realization takes a lot of time for the author to carry out the simulations and to program them in software.

To improve the article, I consider that you should read all the text again to remove small errors, either formatting or grammatical, some examples: spacing between words “ofTreatment” (line 1), repeated words “leads to to” (line 12), “an useful” (line 1) should be replaced by “a useful”, “this effects” (line 8) should be replaced by “these effects”, among others that should be reviewed.

Abstract

For the confidence intervals shown in the abstract, replace “-“ with a comma separating the lower and upper limits of each of the intervals.

Introduction

  • You should clarify the sentence in lines 19 to 22, as citations 2 and 6 are from 2008 and 2015 respectively, so not in the last 5 years!
  • In practical terms, what studies have been carried out in recent years that apply the methods compared in this study?

Materials and Methods

In general, in this section, there are very few citations. Many of the descriptions/models/values already exist in the literature put some references.

  • It remains to specify what some of the indices (j, k, ua, multi, mach) and the variable U mean. Why is it that in section 2.5 the variable Y has an index mach and in sections 2.3 and 2.4 it is worked without an index? Must standardize!
  • In line 67 “s1 – sk” refers to all weights associated with the variable B_zk, I think it is better to denote by “s1,…, sk”.
  • It should further clarify the Monte Carlo simulation section so that a researcher can replicate the work. Lines 79, 82, 83 and 84 are not very clear.
  • Variables C and D do not appear in the model presented in section 2.1, they only appear in the model in section 2.4, I believe that their explanation is better understood in section 2.4 instead of presenting them in section 2.1.
  • In equation (8) the variables u_multi and v_multi appear, which are also weights, but this is not specified. Do these new weights also vary between -2 and 2? Why these values? Is there any theoretical support?
  • On lines 109 and 110, the abbreviation RTEE appears, which must be RTE.
  • In Table 1 you should see the final line. I also suggest specifying a column for the mean (M) and another for the confidence interval (CI). If the CI is in a single line it is easier to read the table. In Table 2 you could also specify the two columns I mentioned. To simplify the two tables, a suggestion would be to eliminate the columns of the p-values, as they are always < 0.001 and, in my opinion, it is enough to put an indication in the text (when talking about Tables 1 and 2) stating that the tests applied presented statistically significant differences (p < 0.001).
  • Reduce/simplify the title of tables and figures (In Instructions for Authors read: All Figures and Tables should have a short explanatory title and caption).

Results

Line 135, replace “RMM” with “MRM”.

Discussion

This section assumes the discussion of the results interpreting them in connection with other studies in the literature. But, it only makes the connection with two of the studies presented in the literature review, so I believe it should be improved by linking to more studies and improving the implications (theoretical and practical) of the study and possible future studies to be carried out.

Good luck with your investigations

Author Response

Abstract

For the confidence intervals shown in the abstract, replace “-“ with a comma separating the lower and upper limits of each of the intervals.

It all has been corrected and highlighted in the text.

Introduction

You should clarify the sentence in lines 19 to 22, as citations 2 and 6 are from 2008 and 2015 respectively, so not in the last 5 years! In practical terms, what studies have been carried out in recent years that apply the methods compared in this study?

The bibliography has been updated and the 5 years expression removed.

Materials and Methods

In general, in this section, there are very few citations. Many of the descriptions/models/values already exist in the literature put some references. It remains to specify what some of the indices (j, k, ua, multi, mach) and the variable U mean. Why is it that in section 2.5 the variable Y has an index mach and in sections 2.3 and 2.4 it is worked without an index? Must standardize!

All mentioned indices have been specified. The U variable is the error derived from the effects not measured by the model (it has been also clarified in the text). The index for Y variables has been added in all equations.

In line 67 “s1 – sk” refers to all weights associated with the variable B_zk, I think it is better to denote by “s1,…, sk”.

It has been corrected

It should further clarify the Monte Carlo simulation section so that a researcher can replicate the work. Lines 79, 82, 83 and 84 are not very clear.

The text has been cleared.

Variables C and D do not appear in the model presented in section 2.1, they only appear in the model in section 2.4, I believe that their explanation is better understood in section 2.4 instead of presenting them in section 2.1.

Variables C and D already appear in section 2.1. A breve reference to their meaning has been added in section 2.4.

In equation (8) the variables u_multi and v_multi appear, which are also weights, but this is not specified. Do these new weights also vary between -2 and 2? Why these values? Is there any theoretical support?

They have been clarified in the text.

On lines 109 and 110, the abbreviation RTEE appears, which must be RTE.

It has been corrected.

In Table 1 you should see the final line. I also suggest specifying a column for the mean (M) and another for the confidence interval (CI). If the CI is in a single line it is easier to read the table. In Table 2 you could also specify the two columns I mentioned. To simplify the two tables, a suggestion would be to eliminate the columns of the p-values, as they are always < 0.001 and, in my opinion, it is enough to put an indication in the text (when talking about Tables 1 and 2) stating that the tests applied presented statistically significant differences (p < 0.001). Reduce/simplify the title of tables and figures (In Instructions for Authors read: All Figures and Tables should have a short explanatory title and caption).

Both tables have been corrected according to reviewer recommendations.

Results

Line 135, replace “RMM” with “MRM”.

Replaced.

Discussion

This section assumes the discussion of the results interpreting them in connection with other studies in the literature. But, it only makes the connection with two of the studies presented in the literature review, so I believe it should be improved by linking to more studies and improving the implications (theoretical and practical) of the study and possible future studies to be carried out.

Good luck with your investigations

Thanks and I really appreciate your kindly recommendations.

Reviewer 2 Report

In the paper,  the behaviors of the Propensity ScoreMatching (PSM) and Multivariate Regression Model (MRM) were analyzed . Monte Carlo Simulations are made to construct groups with different effects in order to compare the behavior of PSM and MRM estimating this effects.  
Some useful results have been achieved.There are still some problems that need to be paid attention to and improved, such as:
  (1) In Section  2.2. Monte Carlo Simulations , why B( n,ρ ) equals to  B( 1,ρz )?   There is no explanation about it in the text。

(2) In Section 2.3. Unadjusted Model, What is the meaning of U in Formula (6) , which is not explained in the text? 

(3) In Section 2.4. Multivariate Regression Model, What is the meaning of D in Formula (8) , which is not explained in the text? 

(4) In Section  2.6. Statistical Analysis,  the input data set of simulation analysis is not provided, and the relevant parameter values about Formula (2),(3),...,(9) are not also  provided,  which leads to poor readability of the article. 
  (5)  The implementation process of PSM algorithm in R language is not explained.

  (6)   Is there any correlation between the data expressed in Figure 1 and Table 1? If so, can Table 1 explain why the trend in Figure 1 is like this?

(7)Is there any correlation between the data expressed in Figure 2 and Table 2? If so, can Table 2 explain  why the trend in Figure 2 is like this?

Author Response

  (1) In Section  2.2. Monte Carlo Simulations , why B( n,ρ ) equals to  B( 1,ρz )?   There is no explanation about it in the text

It has been clarified. What it really means is that a binomial distribution with n=1 and p=pz was used.

(2) In Section 2.3. Unadjusted Model, What is the meaning of U in Formula (6) , which is not explained in the text?

U stands for the error derived from the effects not measured by the model. It is now explained in the text.

(3) In Section 2.4. Multivariate Regression Model, What is the meaning of D in Formula (8) , which is not explained in the text? 

D stands for variables unrelated to treatment and outcomes. It has been clarified in the text.

(4) In Section  2.6. Statistical Analysis,  the input data set of simulation analysis is not provided, and the relevant parameter values about Formula (2),(3),...,(9) are not also  provided,  which leads to poor readability of the article. 

All parameter have been now included and explained in the text.

  (5)  The implementation process of PSM algorithm in R language is not explained.

The PSM algorithm used is the one included in the MatchIt library. The reference has been included in the text.

  (6)   Is there any correlation between the data expressed in Figure 1 and Table 1? If so, can Table 1 explain why the trend in Figure 1 is like this?

Figure 1 represent the data from the third column of table 1. As described in the text it can be seen a major bias reduction with the multivariate model than with the propensity score matching model. These differences increase as real effect increases.

(7)Is there any correlation between the data expressed in Figure 2 and Table 2? If so, can Table 2 explain  why the trend in Figure 2 is like this?

Figure 2 are data from the first 3 columns of Table 2. The trend in the figure is increase in estimated effect, parallel to the real effect. Moreover, as described in the text, a closest estimation to the real effect can be seen with the multivariate model, compared with the others.

Round 2

Reviewer 1 Report

I wish you continued success in your investigations.